# COX16 promotes COX2 metallation and assembly during respiratory complex IV biogenesis

**Abhishek Aich[1], Cong Wang[1], Arpita Chowdhury[1], Christin Ronsör[1], David Pacheu-Grau[1], Ricarda Richter-Dennerlein[1], Sven Dennerlein[1], Peter Rehling[1,2]\***

[1]Department of Cellular Biochemistry, University Medical Centre Göttingen, Göttingen, Germany; [2]Max Planck Institute for Biophysical Chemistry, Göttingen, Germany

**Abstract** Cytochrome *c* oxidase of the mitochondrial oxidative phosphorylation system reduces molecular oxygen with redox equivalent-derived electrons. The conserved mitochondrial-encoded COX1- and COX2-subunits are the heme- and copper-center containing core subunits that catalyze water formation. COX1 and COX2 initially follow independent biogenesis pathways creating assembly modules with subunit-specific, chaperone-like assembly factors that assist in redox centers formation. Here, we find that COX16, a protein required for cytochrome *c* oxidase assembly, interacts specifically with newly synthesized COX2 and its copper center-forming metallochaperones SCO1, SCO2, and COA6. The recruitment of SCO1 to the COX2-module is COX16- dependent and patient-mimicking mutations in SCO1 affect interaction with COX16. These findings implicate COX16 in $Cu_A$-site formation. Surprisingly, COX16 is also found in COX1-containing assembly intermediates and COX2 recruitment to COX1. We conclude that COX16 participates in merging the COX1 and COX2 assembly lines.
DOI: https://doi.org/10.7554/eLife.32572.001

**\*For correspondence:**
peter.rehling@medizin.uni-goettingen.de

**Competing interests:** The authors declare that no competing interests exist.

## Introduction

Aerobic organisms preferentially produce ATP through oxidative phosphorylation. The respiratory chain generates the proton gradient across the inner membrane that drives ATP production by the $F_1F_o$ ATP-synthase. The oxidative phosphorylation system localizes to the inner membrane of mitochondria and comprises of five multi-subunit protein complexes, termed complexes I to V. With the exception of complex II, all complexes are comprised of nuclear- and mitochondrial-encoded subunits. Nuclear-encoded subunits are translated in the cytosol and imported into mitochondria, while the mitochondrial-encoded proteins are translated by membrane–associated mitochondrial ribosomes that associate with the inner membrane for cotranslational protein insertion into the mitochondrial inner membrane. Cytochrome *c* oxidase (COX) is the terminal protein complex of the electron transport chain. COX1, COX2 and COX3 are mitochondrial-encoded subunits that form the core of the complex to which nuclear-encoded proteins associate. The formation and biogenesis process of this complex requires a plethora of chaperone-like factors, termed assembly factors (*Dennerlein and Rehling, 2015*; *Ghezzi and Zeviani, 2012*). Malfunction of many of these assembly factors and the concurrent defects in the assembly process have been linked to severe human disorders that usually affect tissues with high-energy demands, such as neurons, skeletal, and cardiac muscle (*Carlson et al., 2003*; *Fernández-Vizarra et al., 2002*; *Ghezzi and Zeviani, 2012*; *Gorman et al., 2016*).

Assembly of the cytochrome *c* oxidase complex initiates with the synthesis and membrane integration of COX1. Subsequently, imported nuclear-encoded subunits and COX2 and COX3, associate with the COX1-containing assembly module in a sequential manner. Mitochondrial ribosomes selectively translating COX1 mRNA initially associates with the early assembly factor C12ORF62 (hCOX14), MITRAC12 (hCOA3) and CMC1 forming an assembly intermediate termed MITRAC (*Bourens and Barrientos, 2017*; *Carlson et al., 2003*; *Mick et al., 2012*; *Ostergaard et al., 2015*; *Richter-Dennerlein et al., 2016*). MITRAC promotes translation and co-translational membrane insertion of COX1 through association with OXA1L (*Richter-Dennerlein et al., 2016*; *Su and Tzagoloff, 2017*). Furthermore, MITRAC12 provides stability to the newly synthesized COX1 protein. The nuclear-encoded subunits COX4 and COX5A are thought to associate with COX1 prior to recruitment of COX2 into this assembly module (*Dennerlein et al., 2015*; *Mick et al., 2012*).

The assembly process of COX2 initiates with OXA1L- and COX18-mediated membrane insertion (*Fiumera et al., 2007*; *Sacconi et al., 2005*; *Soto et al., 2012*; *Su and Tzagoloff, 2017*). A relay of metallochaperones in the intermembrane space mediates copper insertion into the C-terminus of COX2 and concomitant formation of the $Cu_A$ (*Bourens et al., 2014*; *Carlson et al., 2003*; *Fiumera et al., 2009*; *Khalimonchuk and Winge, 2008*). The copper relay is initiated by COX17, which is crucial for copper delivery to both COX1 and COX2. COX11 mediates the transfer of copper from COX17 to COX1, while delivery of copper to COX2 involves SCO1, SCO2 and COA6 (*Baertling et al., 2015*; *Dennerlein et al., 2015*; *Ghosh et al., 2016*; *Pacheu-Grau et al., 2015*; *Stroud et al., 2015*). A chaperone FAM36A acts in the early steps of COX2 maturation, to provide stability to the newly synthesized protein and act as a scaffold for the metallochaperone (*Bourens et al., 2014*; *Mick et al., 2012*). The copper delivery process is proposed to be sequential, with SCO2 acting upstream of SCO1 (*Baertling et al., 2015*; *Calvo et al., 2012*; *Leary, 2010*; *Stiburek et al., 2009*; *Valnot et al., 2000*). Although the exact mechanism of metalation is still unknown, COA6 appears to cooperate with the SCO proteins in this process. Eventually, COX2 associates with early COX1-containing assembly intermediates and thus the two biogenesis pathways merge.

COX16 is a conserved protein initially identified in yeast as required for the biogenesis of cytochrome *c* oxidase. However, the function of this protein remains ill defined (*Carlson et al., 2003*; *Ghosh et al., 2014*). Based on our finding that COX16 copurifies with assembly intermediates of COX1, we set out to assess the function of COX16 in human mitochondria. As expected, utilizing a human COX16 knock out cell line, we show that COX16 is required for cytochrome *c* oxidase biogenesis. Surprisingly, our analyses demonstrate that COX16 specifically interacts with newly synthesized COX2. In the COX2 biogenesis process, COX16 is required for SCO1 but not SCO2 association with COX2, implicating COX16 in CuA site formation. Patient mimicking amino acid exchanges in SCO1 and COA6 impact COX16 association with these metallochaperones. Moreover, COX16 facilitates COX2 association with the MITRAC assembly intermediate containing COX1. We conclude that COX16 is a constituent of the Copper-insertion machinery and escorts COX2 to the MITRAC-COX1 module for progression of cytochrome *c* oxidase assembly.

## Results

### COX16 interacts with the MITRAC complex

In *S. cerevisiae*, Cox16 has been implicated in the biogenesis of cytochrome *c* oxidase (*Baertling et al., 2015*; *Carlson et al., 2003*). Human COX16 has so far not been analyzed for its function. Recent work on ScCox16 suggested a role in Cox1 biogenesis (*Stiburek et al., 2009*; *Su and Tzagoloff, 2017*). In agreement with this suggestion, we identified human COX16 in affinity purified MITRAC12-containing complexes by quantitative mass spectrometry using stable isotope labeling by amino acids in cell culture (SILAC) (*Mick et al., 2012*; *Valnot et al., 2000*). Accordingly, we identified COX16 among proteins that copurified specifically with COX1-containing assembly intermediates. To confirm the mass spectrometric data, we performed immunoisolation of MITRAC12, C12ORF62 and MITRAC7, all representing assembly factors for COX1 at different stages of the assembly process, from solubilized mitochondria. Established MITRAC components, such as COX1, COX4-1, as well as mitochondrial ribosomes, were efficiently copurified with the baits (*Figure 1A*). However, we observed that COX16 copuridfied solely with MITRAC12. Thus, we

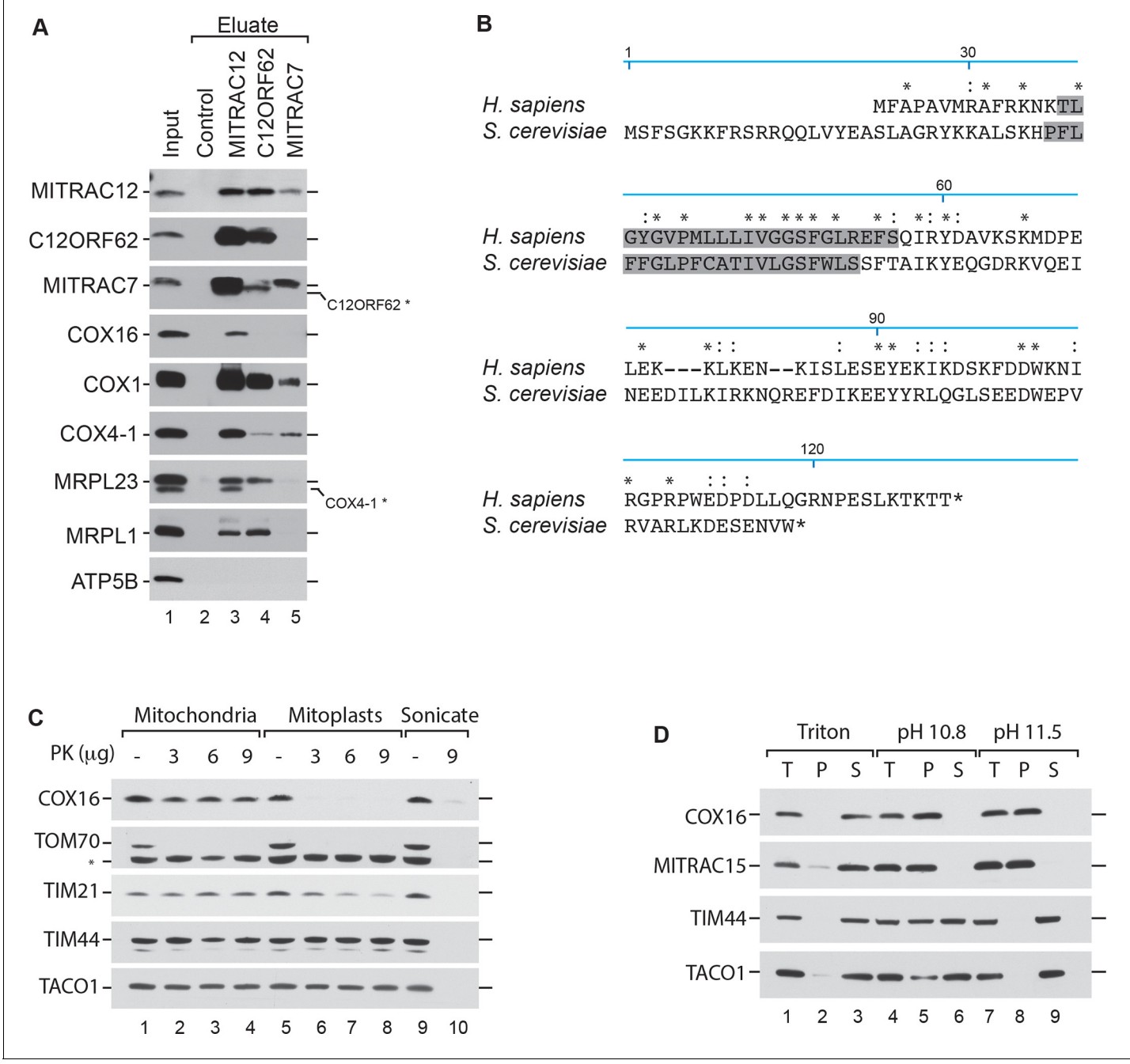

**Figure 1.** COX16 interacts with the MITRAC complex. (A) Antibodies against MITRAC12, C12ORF62, MITRAC7, or control antisera were used for immunoisolations of wild type HEK-293T mitochondria. Total, 3%; Eluates, 100%. Residual signals as a result of redeorations are marked in *. (B) Alignment of the human (*H. sapiens*) COX16 amino acid sequence to its yeast (*S. cerevisiae*) homolog using ClustalW. Predicted transmembrane spans are highlighted in gray; asterisk (*) indicates similar residues; colon (:) indicates identical residues. (C) Submitochondrial localization analysis of COX16 using protease protection assays. Wild type and hypotonically swollen mitochondria were treated with proteinase K (PK). Samples were analyzed by SDS-PAGE and western blotting. Asterisk (*), non-specific signal. (D) Membrane association of COX16 was analyzed using mitochondria that were subjected to carbonate extraction or detergent lysis. T, total; S, soluble fraction; P, pellet.

DOI: https://doi.org/10.7554/eLife.32572.002

confirmed the mass spectrometric data but revealed that COX16 apparently only associates with a specific COX1 assembly state and is not a constitutive COX1-associated factor.

In *Saccharomyces cerevisiae*, Cox16 is an integral innermembrane protein with a mitochondrial targeting sequence at the N-terminus (*Kim et al., 2012*; *Su and Tzagoloff, 2017*; *Tiranti et al., 1999*). However, a comparison between the yeast and human primary sequence showed that human COX16 lacks a predictable Nterminal presequence (*Figure 1B*). Since, the human COX16 does not complement the yeast mutant strain (*Carlson et al., 2003*) we tested the submitochondrial localization of COX16. To this end, we performed hypo-osmotic swelling and carbonate extraction experiments. The recovery of COX16 in each sample was determined by western blot analysis using an antiserum directed against the C-terminus of the protein. COX16 was present in isolated mitochondria and only became accessible to protease treatment when the outer membrane was disrupted (*Figure 1C*). Since COX16 was resistant to carbonate extraction (*Figure 1D*) and has a predicted single transmembrane span, we concluded that Cox16 is an inner mitochondrial membrane protein with its C-terminus facing the intermembrane space (IMS).

## COX16 is required for cytochrome *c* oxidase biogenesis

To assess the function of COX16, we generated a knockout of COX16 using CRISPR/Cas9-mediated disruption of both the alleles in HEK-293T cells. The first exon of the gene was targeted and the selected knockout clone was a compound heterozygote for 26-nucleotide deletion (16_42del) and a single nucleotide insertion (16_17insT). A premature stop codon was introduced in all alleles as a result of the mutations, leading to a complete lack of COX16 in these cells. We carried out steady state analyses of mitochondrial proteins isolated from wild type and COX16$^{-/-}$ cells. COX16$^{-/-}$ mitochondria displayed a marked reduction in the levels of the mitochondrially-encoded COX subunit COX2 and late assembling nuclear-encoded subunits such as COX6C (*Figure 2A*). In the course of our analyses, we also noted that the amount of FAM36A in mitochondria was varied between different mitochondrial preparations. Based on the comparison of several experiments, the slight reduction of FAM36A seen here did not appear to be linked to the presence or absence of COX16. In agreement with the observed reduction of COX proteins in the absence of COX16, COX16$^{-/-}$ cells also displayed growth retardation on glucose- and galactose-containing media (*Figure 2B*).

When we analyzed mitochondrial protein complexes by Blue Native-PAGE (BN-PAGE) and western blotting, we observed that mature cytochrome *c* oxidase was drastically reduced in the COX16$^{-/-}$ cells. As a loading control, the membranes were probed for VDAC. It is interesting to note that most of the COX1 in COX16$^{-/-}$ cells comigrated with the MITRAC assembly intermediate complex (*Figure 2C*). Moreover, we separated solubilized respiratory chain complexes by BN-PAGE and performed in-gel activity staining for complexes I, IV, and V. While the activities of complex V were similar between wild type and COX16 knockout mitochondria, complex IV activity appeared significantly reduced and complex I activity slightly increased at the level of the supercomplex (*Figure 2D*). When mitochondria were solubilized in DDM to dissociate supercomplexes, we observed that the absolute amount of mature cytochrome *c* oxidase was drastically affted in the COX16$^{-/-}$ cells and that COX1 was mainly present in faster migrating complexes (*Figure 2E*). For a quantitative assessment, we measured cytochrome *c* oxidase activity and quantified the amount of enzyme by ELISA. In COX16 knockout cells, the cytochrome *c* oxidase amount was reduced to ~50% compared to the wild type control (*Figure 2F*, left). This reduction directly correlated with the reduction of complex IV activity to ~65%, as compared to the control (*Figure 2F*, right). In view of all these results, we concluded that loss of COX16 in HEK-293T cells lead to a severe reduction of cytochrome *c* oxidase.

## COX16 is required for COX2 assembly

To analyze the effect of absence of COX16 on mitochondrial protein synthesis, mitochondrial translation products were pulsed labeled with [$^{35}$S]methionine. However, no significant differences were observed in the synthesis of either COX1, COX2, or any other protein (*Figure 3A*). This was further supported by quantifications of the labeled proteins. Since protein synthesis was apparently not affected, we addressed whether the reduction of COX1 or COX2 in COX16$^{-/-}$ cells was a result of reduced protein stability. Pulse chase analysis was carried out to follow the fate of newly synthesized COX1 or COX2 over the course of 24 hr (*Figure 3B*). (Surprisingly, we observed enhanced synthesis

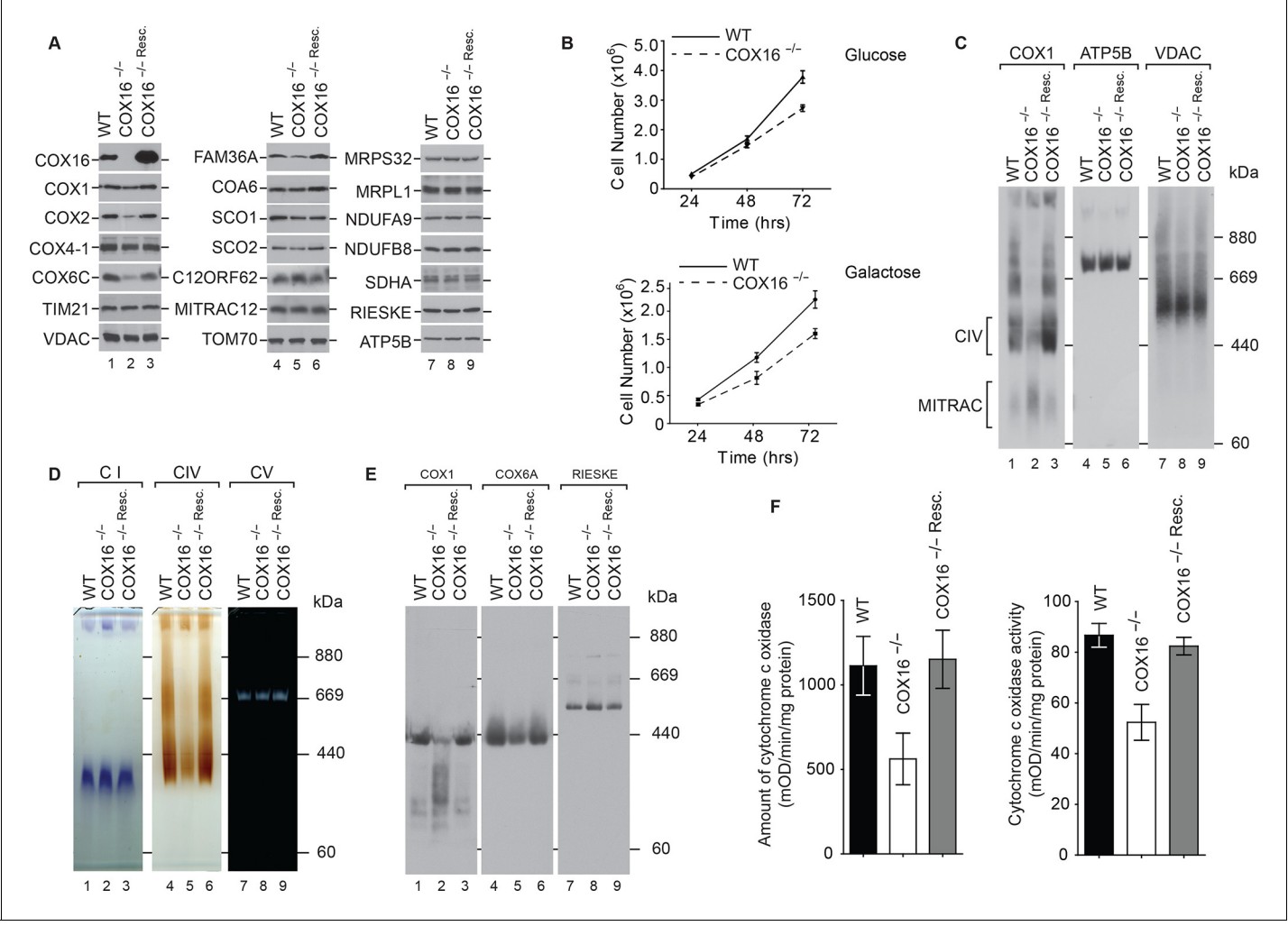

**Figure 2.** COX16 is required for cytochrome *c* oxidase biogenesis. (**A**) Isolated wild-type (WT), COX16 knockout (COX16[-/-]) and COX16 knockout expressing WT COX16 from the T-REx locus (COX16[-/- Resc.]) mitochondria were analyzed by western blotting. (**B**) Cell count of wild-type (WT) and COX16 knockout (COX16[-/-]) grown in medium containing either glucose (left) or galactose (right). (**C**) Isolated mitochondria from (**A**), solubilized in 1% Digitonin and analyzed by BN-PAGE and western blotting, subjected with antisera against COX1, ATP5B and VDAC. CIV, complex IV. (**D**) Isolated mitochondria from (**A**), solubilized in 1% Digitonin and analyzed by BN-PAGE and in-gel activity assays for complex I (CI), IV (CIV) and V (CV). (**E**) Isolated mitochondria from (**A**), were solubilized in 1% N-Dodecyl β-D-maltoside (DDM) and analyzed by BN-PAGE and western blotting with indicated antisera. (**F**) Measurement of relative amount of cytochrome *c* oxidase (right) and enzyme activity (left); the mitochondria were isolated as mentioned in (**A**) (mean ±SEM and n = 3).

DOI: https://doi.org/10.7554/eLife.32572.003

and stability of ATP6 and ATP8 in the absence of COX16.) Newly synthesized subunits are usually incorporated into the mature complex in the chosen time frame (*Dennerlein et al., 2015*; *Leary et al., 2009*). Interestingly, COX2 showed a marked reduction in stability in the absence of COX16 (*Figure 3C*). A similar but non significant reduction was observed for COX1; however, proteins such as ND3 remained unaffected.

To address whether the observed reduction in COX2 amounts correlated with a defect in the assembly process, labeled mitochondria-encoded proteins that assembled into mitochondrial protein complexes were analyzed by BN-PAGE followed by separation on the second dimension using SDS-PAGE. These analyses revealed a notable reduction of newly synthesized COX2 in mature cytochrome *c* oxidase (*Figure 3D*). To further analyze mitochondrial protein complexes with respect to the presence of nuclear-encoded proteins, mitochondria from wild type and COX16[-/-] cells were analysed by 2D-BN/SDS-PAGE and western blotting. In agreement with our previous result, the

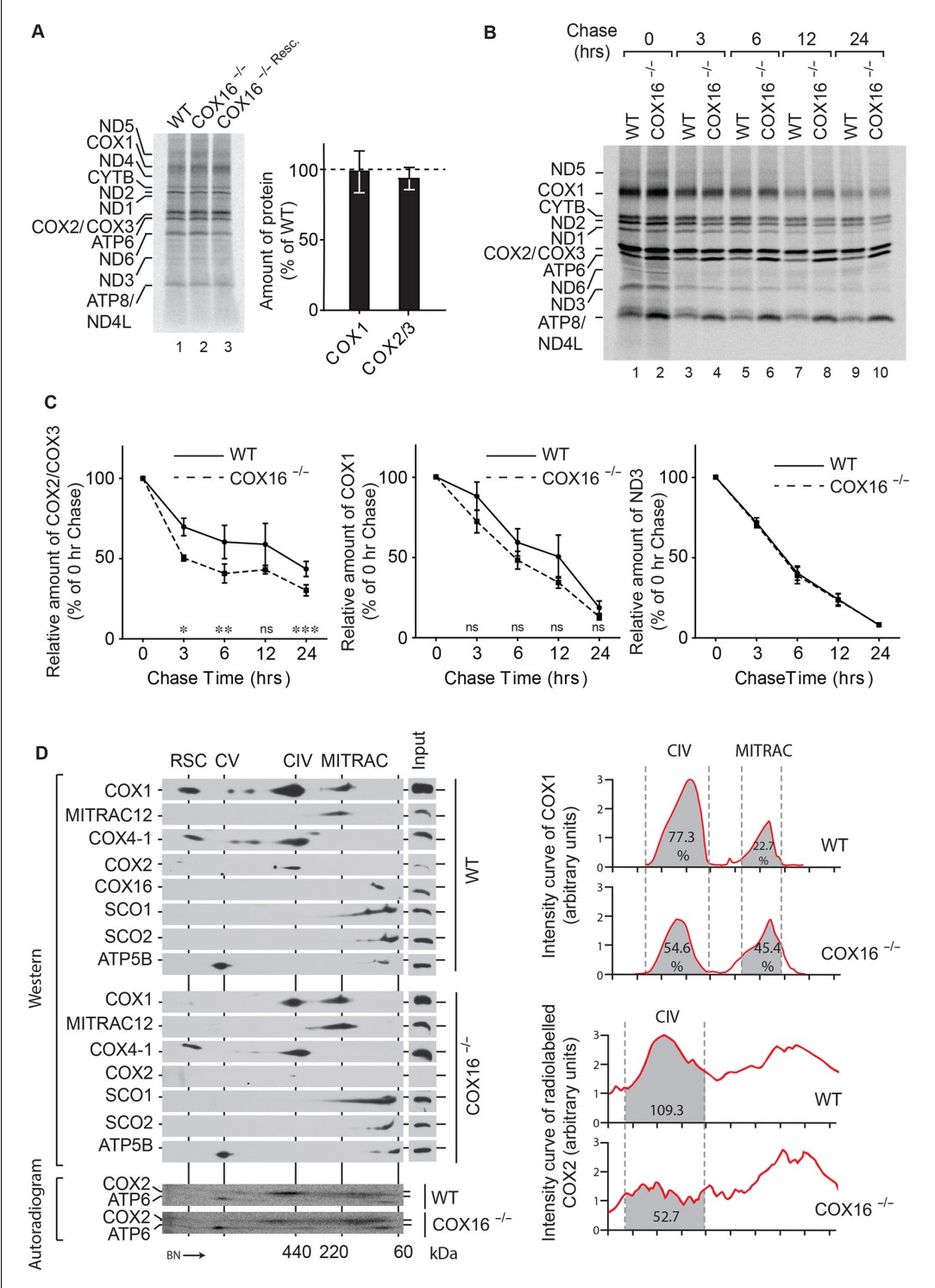

**Figure 3.** COX16 is required for COX2 assembly. (**A**) In vivo labeling of mitochondrial translation products with [$^{35}$S]methionine in wild-type (WT), COX16 knockout (COX16$^{-/-}$) and COX16 knockout expressing WT COX16 from the T-REx locus (COX16$^{-/-}$ $^{Resc}$.). Cells were pulsed for 1 hr and analyzed by SDS-PAGE and digital autoradiography. The values represented are quantifications of the indicated mitochondrial translation products normalized to ND1 (mean ± SEM and n = 3). (**B**) Mitochondrial translation products in wild-type (WT) and COX16 knockout (COX16$^{-/-}$) were labeled with [35S] methionine for 1 hr. Subsequently, the medium was replaced and cells were further cultured in standard medium (chase) for 3, 6, 12 and 24 hr. Cell extracts were analyzed by SDS-PAGE and digital autoradiography. (**C**) Quantifications using ImageQuant software of the indicated mitochondrial translation products from (B). The values represented were normalized to ND1 (mean ± SEM and n = 3; *p=0.029, **p=0.042, ***p=0.024, ns = non

*Figure 3 continued on next page*

*Figure 3 continued*

significant). (D) Protein complexes from wild-type (WT) and COX16 knockout (COX16$^{-/-}$) mitochondria were extracted under non-denaturing conditions and separated by BN-PAGE, followed by a second dimension SDS-PAGE and western blot analysis (top). Mitochondrial translation products were labeled with [$^{35}$S]methionine, prior whole cell lysis and complexes separation as described above (bottom). The proteins were detected by using indicated antibodies or by digital autoradiography (COX2, ATP6). Intensity curves (right) for COX1 signals from the western blotting and COX2 from the autoradiogram were calculated using ImageJ. Numbers in the gray regions denote area under intensity curves. For COX1 (top), it is represented as percentage of the total signal in CIV and MITRAC and for COX2 (bottom), as arbitrary units. CIV, Monomeric Complex IV; CV, Complex V; RSC, Respiratory Super-Complexes.

DOI: https://doi.org/10.7554/eLife.32572.004

ratios of COX1 present in the mature monomeric protein complex IV and in MITRAC complexes changed significantly in absence of COX16. This finding suggested that COX1 accumulated in MITRAC complexes in these cells. Similarly, COX2 was barely visible in the mature cytochrome *c* oxidase. Based on these findings, we concluded that a lack of COX16 impacts the maturation of COX1-containing assembly intermediates.

## COX16 is required for SCO1 interaction with COX2

To clarify, if COX16 participated in the biogenesis of COX1 directly, we assessed association of COX16 with mitochondrial translation products. Therefore, we performed immunoisolations of COX16 from wild type cells after radiolabeling of mitochondrial translation products. Unexpectedly, in these analyses COX16 solely associated with the newly synthesized COX2 (*Figure 4A*). Accordingly, COX16 is involved in the biogenesis of the COX2 assembly module.

To this end, we addressed if absence of COX16 disturbed the assembly of COX2. Therefore, we performed immunoisolation of the COX2-specific metallochaperones SCO1, SCO2, and COA6 from wild type and COX16$^{-/-}$ mitochondria. COX16 coisolated with SCO1 and COA6 in immunoisolations from wild type cells, indicating that these proteins form a complex (*Figure 4B*). Remarkedly, we did not observe coisolation of COX16 with SCO2. While a lack of COX16 lead to loss of COX2 association with SCO1 and COA6, the association between SCO2 and COX2 remained unaffected.

To address the fate of newly synthesized COX2 in the absence of COX16, we carried out immunoisolations of SCO1, SCO2, COA6, and FAM36A after radiolabeling of mitochondrial translation products. Interestingly, only the association of SCO1 with newly synthesized COX2 was affected in the absence of COX16 (*Figure 4C and D*). The binding of other COX2 chaperones such as SCO2, COA6, and FAM36A was unaffected by the absence of COX16. In contrast to the analysis shown in *Figure 4B*, these immunoprecipitations were performed from whole cells. Under these conditions, we do not observe a SCO2 signal in the COA6 eluates, probably due to the reduced amounts of protein. Based on this, we addressed if the interaction between COX16 and COX2 metallochaperones and the scaffold FAM36A depended on the presence of COX2. To this end, we performed immunoisolation of SCO1, SCO2, COA6, and FAM36A from mitochondria isolated from non-treated cells or cells treated with thiamphenicol, a specific inhibitor of mitochondrial translation (*Banci et al., 2008*; *Mick et al., 2012*). Under conditions of thiamphenicol treatment, associations between COX16 and any of the tested COX2-associated chaperones were lost indicating that newly synthesized mitochondrial-encoded proteins, likely COX2, are required for the interactions (*Figure 4E*). It is crucial to note that thiamphenicol treatment did not affect the steady state levels of COX2, supporting the idea that the loss of interaction observed in the experiment was due to the absence of newly synthesized COX2.

Pathogenic mutations have been reported for both SCO1 and COA6 - the two proteins with which COX16 apparently predominantly associates. SCO1 patients suffer from hypertrophic cardiomyopathy, neonatal hepatopathy, and ketoacidotic comas, whereas mutations in COA6 cause fatal infantile cardioencephalomyopathy (*Baertling et al., 2015*; *Banci et al., 2007*; *Calvo et al., 2012*; *Cobine et al., 2006*; *Stiburek et al., 2009*; *Valnot et al., 2000*). Since the clinical presentation of the patients differed markedly, it was intriguing to assess if COX16-association was selectively disturbed. Therefore, we transiently expressed C-terminally FLAG-tagged COA6 or SCO1 along with variants harboring individual pathogenic substitutions - W59C (*Ghosh et al., 2014*) and W66R (*Baertling et al., 2015*) in case of COA6, G132S (*Stiburek et al., 2009*) and P174L (*Valnot et al., 2000*) in SCO1. We performed immunoisolations of the FLAG-tagged proteins. Interestingly, we

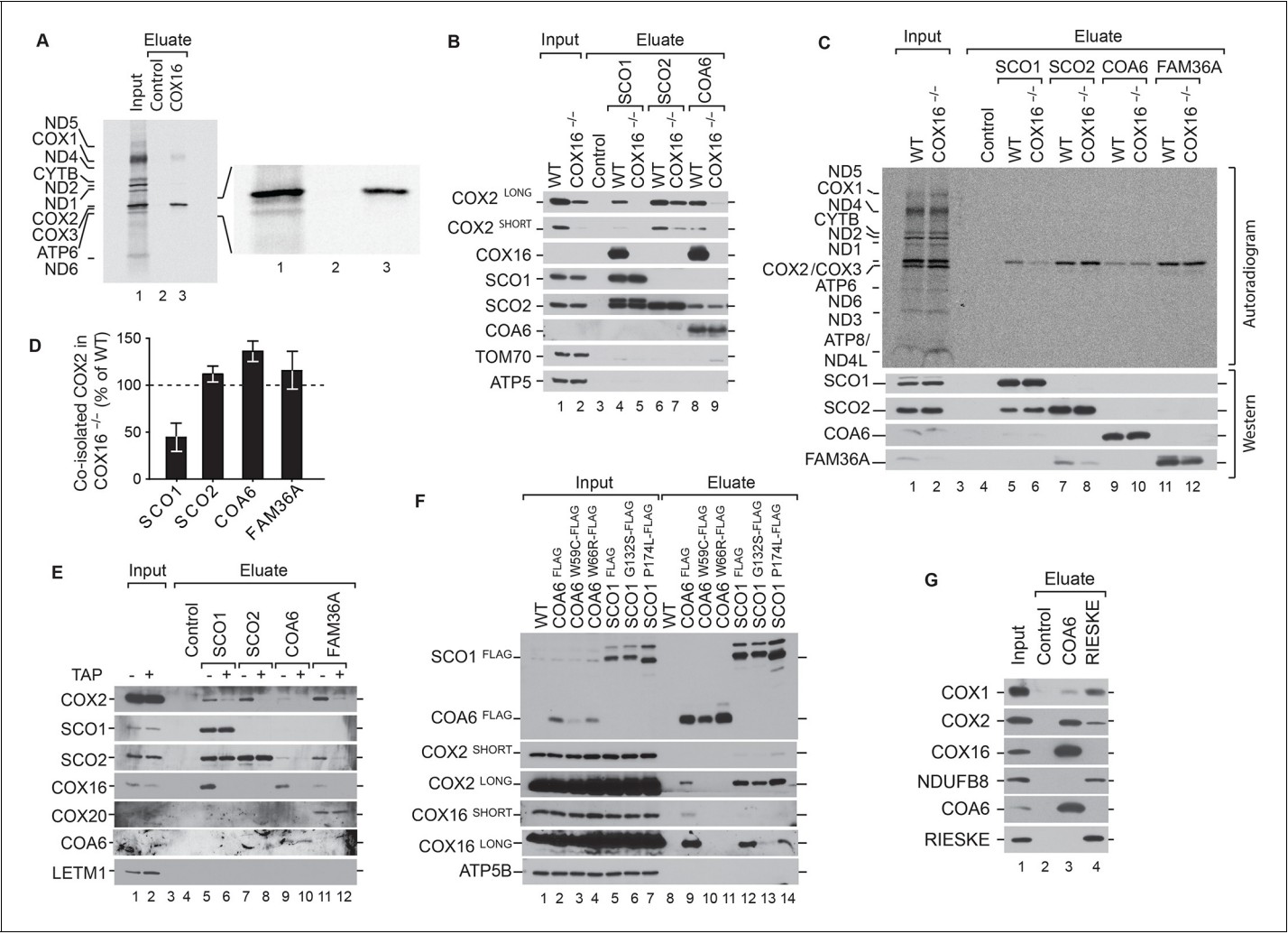

**Figure 4.** COX16 is required for SCO1 interaction with COX2. (**A**) Mitochondrial translation products of wild type cells were labeled with [35S] methionine for 1 hr and whole cell extracts subjected to immunoprecipitation using anti-COX16 or control antisera. The eluates were analyzed by digital autoradiography after SDS-PAGE (Total, 5% and Eluate, 100%). (**B**) Immunoprecipitation from wild-type (WT) and COX16 knockout (COX16$^{-/-}$) mitochondria with anti-SCO1, anti-SCO2, anti-COA6 or control antisera. The eluates were analyzed by western blotting after SDS-PAGE with the indicated antibodies (Total 5% and Eluate, 50%). (**C**) Mitochondrial translation products in wild-type (WT) and COX16 knockout (COX16$^{-/-}$) were labeled with [35S]methionine for 1 hr. Whole cell extracts were subjected to immunoprecipitation with anti-SCO1, anti-SCO2, anti-COA6, anti-FAM36A or control antisera. Eluates were separated by SDS-PAGE and Western blotting. Radioactive signals were visualized by digital autoradiography and membranes afterwards decorated with the indicated antibodies. (Total, 5% and Eluate, 100%). (**D**) Quantification of co-isolated COX2 from immunoprecipitations with the indicated antibodies from (**C**) (mean ± SEM and n = 3). (**E**) Mitochondria (without (-) or with thiamphenicol (TAP) (+) treatment) from wild-type (WT) and COX16 knockout cells (COX16$^{-/-}$) were subjected to immunoprecipitations using antibodies against SCO1, SCO2, COA6 and FAM36A. Samples were subjected to SDS- PAGE and analyzed by western blotting using the indicated antibodies (Total, 5% and Eluate, 100%). (**F**) Immunoisolations of COA6$^{FLAG}$ or SCO1$^{FLAG}$ along with variants harboring individual pathogenic substitutions (COA6 - W59C and W66R, SCO1 - G132S and P174L). Cells were solubilized and subjected to anti-FLAG immunoprecipitation and eluates analyzed by SDS-PAGE and western blotting using the indicated antibodies. WT, wild type. (Total 5% and Eluate, 50%). (**G**) Immunoprecipitation from wild-type (WT) mitochondria with anti-COA6, anti-RIESKE or control antisera. The eluates were analyzed by western blotting after SDS-PAGE with the indicated antibodies (total 5% and eluate, 50%).

DOI: https://doi.org/10.7554/eLife.32572.005

observed a considerable loss of association of COX16 with the pathogenic variants of SCO1. At the same time, the mutant versions of SCO1 maintained their association with COX2 (*Figure 4F*). This finding indicated that the developed pathology in these patients was not due to defective COX2 recruitment but rather lack of association with COX16. In contrast, association of COX16 with pathogenic variants of COA6 was lost with a concomitant loss of interaction with COX2. In summary,

COX16 is part of and directly involved in the biogenesis of the COX2 assembly module. Loss of COX16 association with SCO1 is a hallmark of the tested SCO1 patient models. To assess if COX16 was associated with respiratory chain supercomplexes, we isolated supercomplexes via the RIESKE protein of complex III. RIESKE purification led to coisolation of complex IV and I components. However, COX16 was not copurified in contrast to immunoisolation with the metallochaperone COA6 (*Figure 4G*). We conclude that COX16 does not act on respiratory chain supercomplexes.

## COX16 facilitates integration of COX2 into MITRAC-COX1 modules

Based on the above considerations, we hypothesized that a block in early stages of COX1 assembly may reflect accumulation of intermediates in COX2 assembly, which can not engage with COX1 to promote further maturation steps. To test this directly, we utilized SURF$^{-/-}$ cells (*Kim et al., 2012*; *Tiranti et al., 1999*) and immunoisolated endogenous COX16 and COA6 after radiolabeling of mitochondrial translation products. In both cases, COX2 was specifically enriched in COX16 and COA6 eluates from wild type cells. However, the amount of COX2 was significantly enriched in COX16 and COA6 eluates from SURF$^{-/-}$ cells (*Figure 5A*). Hence, a block in COX1 assembly lead to accumulation of COX2-containing assembly modules containing COX16 or COA6. To substantiate the idea that COX16 associates with the MITRAC complexes only upon association with COX2, we targeted early steps of COX2 assembly. FAM36A is an early chaperone crucial for synthesis of COX2 (*Bourens et al., 2014*). Hence, we performed immunoisolations of MITRAC12 from wildtype and FAM36A$^{-/-}$ mitochondria. Upon loss of COX16, COX2 was no longer copurified with MITRAC12, indicating that COX16 interacted with MITRAC12 only after associating with COX2 (*Figure 5B*). To this end, we compared the amount of COX16, that associated with COX2 pathway constituents to that in complex with MITRAC12. Both COX16 and COX2, associated with MITRAC12 to a significantly lower magnitude as compared to other COX2 associated proteins (*Figure 5C*), indicating that this interaction might be a fairly transient one. We were able to enrich the MITRAC12-COX16 complex when we applied native purification via MITRAC12$^{FLAG}$ (*Figure 5D*). Under these conditions, we were able to also detect COX16 in complex with C12ORF62 (*Figure 5E*), which had not been apparent in the immunosiolations using the C12ORF62 antiserum (*Figure 1A*).

These findings raised the question if COX16 was required for recruitment of COX2 to COX1-containing MITRAC complexes. To this end, we examined protein co-purification with C12ORF62 and MITRAC12 in the presence or absence of COX16. Our analyses showed that COX2-association with MITRAC12 or C12ORF62 was drastically affected by the absence of COX16 (*Figure 5F*). Moreover a slight but consistent increase in the association of COX1 with C12ORF62 and MITRAC12 was apparent. When similar imunoisolation experiments were carried out after radiolabelling of mitochondrial translation products, we observed a specific association of MITRAC12 with newly synthesized COX2, that was not observed for C12ORF62 (*Figure 5G*). Moreover, in the absence of COX16, a defect in association of the newly synthesized COX2 with MITRAC12 was apparent (*Figure 5G*). Concomitantly, the C12ORF62 and MITRAC12-associated COX1 levels were increased (*Figure 5G*). We conclude, that COX16 facilitates the assembly of COX2-containing assembly modules to the COX1-containing MITRAC-complexes possibly together with MITRAC12. However, association of MITRAC12 with COX2 is stimulated by COX16.

## Discussion

In this study, we demonstrate an involvement of COX16 in the formation of the COX2 assembly module. Despite the identification and functional assessment of several assembly factors involved in the process of COX2 metalation, it has remained unknown as to how the COX2-containing module engages with its partner subunit COX1. Here we find that COX16 acts at two distinct stages of COX2 biogenesis, the recruitment of metallochaperones to COX2 and the merging COX2 and COX1 assembly routes (*Figure 6*). Our analyses show: (1) COX16 facilitates association of the metallochaperone SCO1 to newly synthesized COX2; (2) the early metalochaperone COA6 strongly associates with COX16, further supporting the role of COX16 in COX2 maturation; (3) COX16 acts at a checkpoint for proper COX2 maturation leading to increased turnover of COX2 in the absence of COX16; (4) the COX2 assembly module accumulates in the absence of COX1 hemelation (SURF knockdown); (5) a small amount of COX16 also interacts with COX1-containing intermediates (copurified by C12ORF62 and MITRAC12) in an apparently transient manner; and (6) COX16 promotes

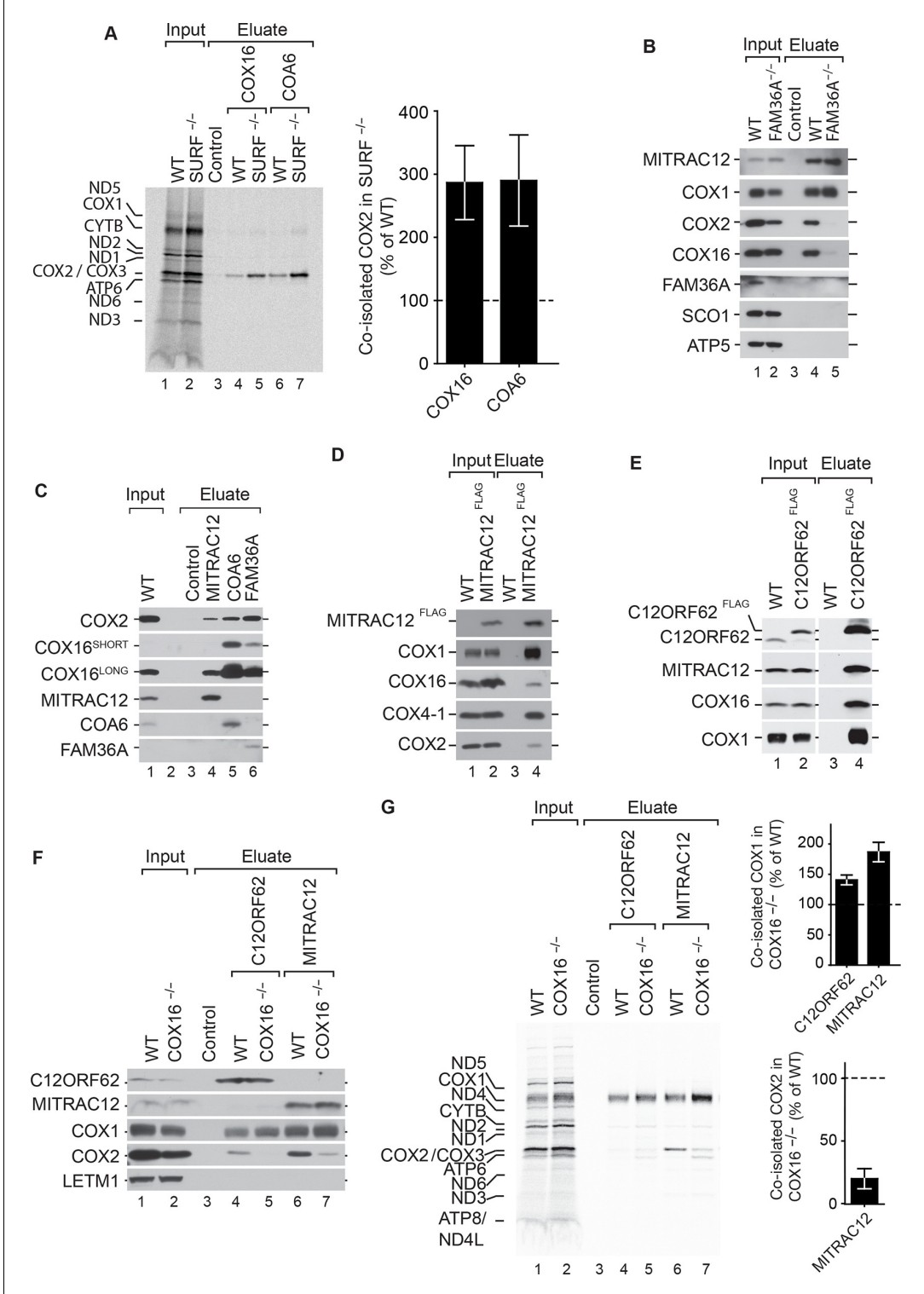

**Figure 5.** COX16 facilitates integration of COX2 into MITRAC-COX1 modules. (**A**) Mitochondrial translation products in wild-type (WT) and SURF knockout (SURF$^{-/-}$) were labeled with [$^{35}$S]methionine for 1 hr. Whole cell extracts were subjected to immunoprecipitation using anti-COX16, anti-COA6 or control antisera. Eluates were analyzed by digital autoradiography after SDS-PAGE (Total, 5% and Eluate, 100%). Quantification of co-isolated COX2 amounts with the indicated antibodies were performed using ImageJ (mean ± SEM and n = 3). (**B**) Immunoprecipitation from wild-type (WT) and FAM36A knockout (FAM36A$^{-/-}$) mitochondria with anti-MITRAC12 or control antisera. The eluates were analyzed by western blotting after SDS-PAGE with the indicated antibodies (Total 5% and Eluate, 100%). (**C**) Immunoprecipitation from wild-type (WT) mitochondria with anti-MITRAC12, anti-COA6, anti-FAM36A or control antisera. The eluates were analyzed by western blotting after SDS-PAGE with the indicated antibodies (Total 5% and Eluate,

*Figure 5 continued on next page*

*Figure 5 continued*

50%). (D) Mitochondria isolated from induced MITRAC12[FLAG] cells were solubilized and subjected to anti-FLAG immunoprecipitation and eluates analyzed by SDS-PAGE and western blotting using the indicated antibodies. WT, wild type. (Total 5% and Eluate, 50%). (E) Mitochondria isolated from induced C12ORF62[FLAG] cells were solubilized and subjected to anti-FLAG immunoprecipitation and eluates analyzed by SDS-PAGE and western blotting using the indicated antibodies. WT, wild type. (Total 3% and Eluate, 100%). (F) Mitochondria isolated from wild-type (WT) and COX16 knockout (COX16[−/−]) were used for immunoprecipitation with anti-MITRAC12 or control antisera. The eluates were analyzed by western blotting after SDS-PAGE with the indicated antibodies (Total 5% and Eluate, 50%). (G) Antibodies against C12ORF62, MITRAC12 or control antisera were used for immunoisolation after [$^{35}$S]methionine labeling of mitochondrial translation products in wild-type (WT) and COX16 knockout (COX16[−/−]) cells and analyzed by SDS-PAGE and digital autoradiography (Total, 5% and Eluate, 100%). Quantification of co-isolated COX1 or COX2 amounts with the indicated antibodies were performed using ImageJ (mean ± SEM and n = 3).

DOI: https://doi.org/10.7554/eLife.32572.006

the integration of COX2 into the COX1-assembly module thereby becoming a transient constituent of the above mentioned intermediate.

Cu$_A$ site formation on COX2 requires a relay of copper chaperones. Copper-loaded SCO2 is thought to bind to COX2 either as it is inserted into the inner membrane or immediately thereafter (*Leary et al., 2009*). This promotes SCO1 to be metallated by COX17 (*Banci et al., 2008*). There seems to be a dichotomy in the order of these steps. In the first scenario, both SCO2 and SCO1 sequentially bound to COX2 together, delivering one Cu$^{2+}$ each in the Cu$_A$ site. The other possibility involves their binding being spatiotemporally separated. Here we observe that in the absence of COX16 there is a reduction in the ability of newly synthesized COX2 to associate with SCO1 but not SCO2 (*Figure 4C*). Thus, irrespective of whichever integration model for the SCO proteins be correct, COX16 specifically mediates the association SCO1 with COX2. Both scenarios have another layer of complexity in terms of Cu$^{2+}$-ion association with SCO1. We observe that mutations in SCO1 do not reduce its ability to bind to COX2 (*Figure 4D*). However, the association with COX16 is severely affected in the mutants. Studies on the P$^{174}$L mutant, led to the idea that SCO1 has distinct faces for interaction with COX17 and COX2 (*Banci et al., 2007*; *Cobine et al., 2006*). Thus, it is

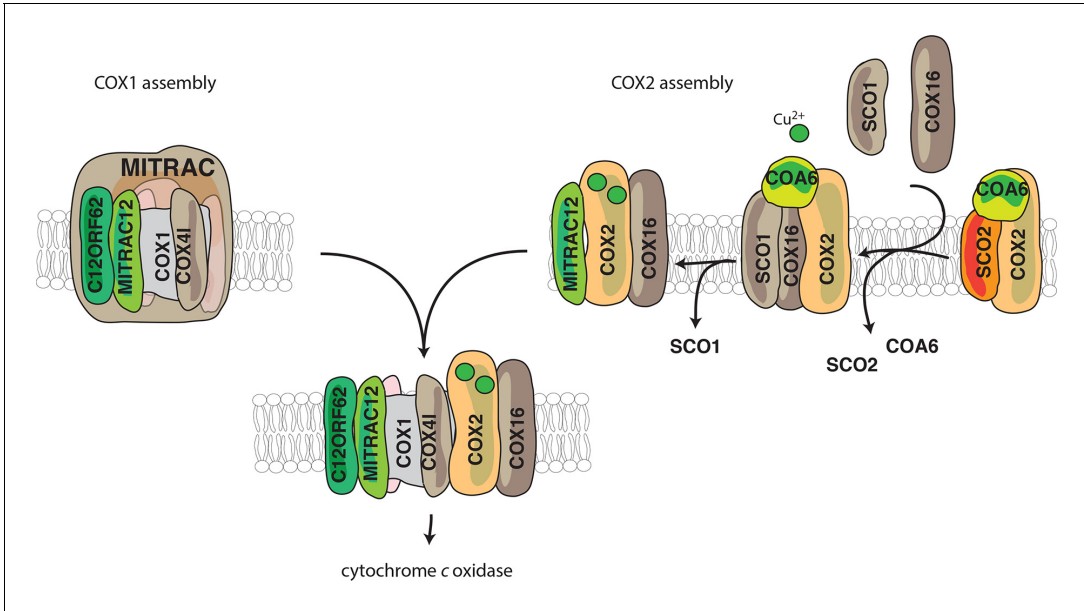

**Figure 6.** Model for the role of COX16. COX1 is assembled and guided through the assembly process through its association with MITRAC, where it awaits the association of COX2. COX2 is initially associated with FAM36A and metallochaperones such as SCO2 and COA6 in the early assembly stages. COX16 acts at the later stages of COX2 assembly. The early assembly factors are apparently no longer associated with the complex at this stage. COX16 facilitates the association of SCO1 and thus probably leads to proper COX2 maturation. It then facilitates the merger of COX1 and COX2 assembly lines after the exit of SCO1.

DOI: https://doi.org/10.7554/eLife.32572.007

conceivable that COX17 delivers $Cu^{2+}$ to SCO1 only after SCO1 is bound to COX2. Hence, based on our studies, we propose that COX16 facilitates SCO1-COX2 binding and potentially the subsequent interaction of SCO1 with COX17. In agreement with a SCO1-specific function of COX16, the tested COA6 mutants loose association to COX2 and concomitantly also to COX16 (*Figure 4D*). This indicates that events leading to lack of interaction of COX16 with SCO1 mutants are different to that of COA6 mutants.

During assembly of the cytochrome *c* oxidase, particularly in the COX1 assembly module, MITRAC7 stabilizes COX1 in a late MITRAC intermediate (*Dennerlein and Rehling, 2015*; *Dennerlein et al., 2015*; *Ghezzi and Zeviani, 2012*). This is considered to enable the COX1-containing complex to receive additional subunits. Since absence of COX16 does not allow the assembly of COX2 to proceed, we find an accumulation of COX1 on MITRAC12 and C12ORF62 both being involved in initial stages of COX1- translation and assembly (*Figure 5C* and *Figure 5D*). Similarly, by creating a block in the maturation of the COX1 module in SURF knockout cells, the COX2 assembly module accumulated together with COX16 (*Figure 5A*). This supports current models according to which both assembly modules initiate independent of each other (*Fernández-Vizarra et al., 2002*; *Ghezzi and Zeviani, 2012*; *Gorman et al., 2016*; *McStay et al., 2013*; *Timón-Gómez et al., 2017*). Since, COX16 and COX2 are co-dependent with regard to their interaction with MITRAC12 (*Figure 5B* and *Figure 5C*), we suggest that COX16 acts initially in COX2 maturation. Furthermore, we observe that only COX16 but not SCO1 associate with MITRAC12 (*Figure 5B*). This finding suggests that the function of COX16 extends beyond SCO1 recruitment and that the role in merging the assembly lines between COX1 and COX2 is downstream of SCO1-dependent metalation. Hence, it is tempting to speculate that COX16 recognizes readiness for merging COX2 into the COX1 assembly line. In addition, we observe that MITRAC12, which is required for COX1 biogenesis, also binds to newly synthesized COX2 in a COX16-dependent manner. In contrast, C12ORF62, the most early COX1 interacting protein, does not display association with newly synthesized COX2. Hence, it is conceivable, that MITRAC12 cooperates with COX12 in late steps of COX2 biogenesis to link it to the COX1 module (*Figure 6*).

The finding of an interaction between human COX16 and COX1-containing assembly interemediates is in line with a recent study in yeast (*Bourens and Barrientos, 2017*; *Mick et al., 2012*; *Ostergaard et al., 2015*; *Richter-Dennerlein et al., 2016*; *Su and Tzagoloff, 2017*). However, the observed presence of yeast Cox16p in mature cytochrome *c* oxidase and its supercomplexes is not conserved in human. In fact, this finding illustrates that the situation in human cells differs considerably from yeast. We demonstrate that human COX16 is primarily associated with COX2 assembly modules, which has not been investigated in yeast. The association of COX16 with MITRAC assembly intermediates containing COX1 is COX2-dependent and thus likely transient in nature. Importantly, BN-PAGE analyses demonstrated that COX16 co-migrates with SCO1-containing protein complexes supporting quantitative association of COX16 with COX2 assembly intermediates (*Figure 3D*). In summary, we demonstrate that human COX16 functions to stabilize the interaction between SCO1 and newly synthesized COX2. It subsequently facilitates the maturation of COX2 for proper insertion of $Cu^{2+}$ into $Cu_A$ sites of COX2. Lack of COX16 leads to increased turnover of newly synthesized COX2 and accumulation of COX1 in MITRAC. Hence, COX16 cooperates with MITRAC12 and copper chaperones to facilitate COX2 assembly with COX1-containing intermediate.

# Materials and methods

## Key resources table

| Reagent type (species) or resource | Designation | Source or reference | Identifiers | Additional information |
|---|---|---|---|---|
| Gene (Homo sapiens) | COX16 | NCBI | NCBI Gene ID: 51241 | First study to adress Human COX16 |
| Cell line (Homo sapiens) | HEK293-Flp-InTM T-RexTM (HEK293T) Cell Line | ThermoFisher Scientific | RRID:CVCL_U421 | |
| Cell line (Homo sapiens) | HEK293-Flp-InTM T-RexTM (HEK293T)-COX16-/- | This paper | N/A | Cell line generated as described in Materials and methods |

*Continued on next page*

*Continued*

| Reagent type (species) or resource | Designation | Source or reference | Identifiers | Additional information |
|---|---|---|---|---|
| Cell line (Homo sapiens) | HEK293-Flp-InTM T-RexTM (HEK293T)-COX16-/- COX16 OE | This paper | N/A | Cell line generated as described in Materials and methods |
| Transfected construct (Homo sapiens) | pX330-COX16 gRNA | This paper | N/A | cloning described in Materials and methods |
| Transfected construct (Homo sapiens) | pEGFPN1 | Clonetech | N/A | |
| Transfected construct (Homo sapiens) | pCDNA5-COX16 | This paper | N/A | Construct generated from amplifying COX16 from WT HEK cDNA |
| Transfected construct (Homo sapiens) | pCDNA5-COA6-FLAG | This paper | N/A | Construct generated from amplifying COA6 from WT HEK cDNA |
| Transfected construct (Homo sapiens) | pCDNA5-COA6-W59C-FLAG | This paper | N/A | Construct generated by mutagenesis of pCDNA5-COA6-FLAG |
| Transfected construct (Homo sapiens) | pCDNA5-COA6-W66R-FLAG | This paper | N/A | Construct generated by mutagenesis of pCDNA5-COA6-FLAG |
| Transfected construct (Homo sapiens) | pCDNA5-SCO1 | This paper | N/A | Construct generated from amplifying SCO1 from WT HEK cDNA |
| Transfected construct (Homo sapiens) | pCDNA5-SCO1-G132S-FLAG | This paper | N/A | Construct generated by mutagenesis of pCDNA5-SCO1-FLAG |
| Transfected construct (Homo sapiens) | pCDNA5-SCO1-P174L-FLAG | This paper | N/A | Construct generated by mutagenesis of pCDNA5-SCO1-FLAG |
| Biological sample () | N/A | | | |
| Antibody | MITRAC12 | self made | PRAB3761 | (1:1000) |
| Antibody | C12ORF62 | self made | PRAB4844 | (1:500) |
| Antibody | MITRAC7 | self made | PRAB4843 | (1:500) |
| Antibody | COX16 | Proteintech | RRID:AB_10666854 | (1:1000) |
| Antibody | COX1 | self made | PRAB2035 | (1:2000) |
| Antibody | COX4-1 | self made | PRAB1522 | (1:2000) |
| Antibody | MRPL23 | self made | PRAB1716 | (1:500) |
| Antibody | MRPL1 | self made | PRAB4969 | (1:500) |
| Antibody | TOM70 | self made | PRAB3280 | (1:1000) |
| Antibody | TACO1 | self made | PRAB3627 | (1:500) |
| Antibody | MITRAC15 | self made | PRAB4814 | (1:500) |
| Antibody | FLAG | Sigma Aldrich | RRID:AB_259529 | (1:2000) |
| Antibody | COX2 | Abcam | ab110258 | (1:2000) |
| Antibody | TIM21 | self made | PRAB3674 | (1:2000) |
| Antibody | VDAC | self made | PRAB1515 | (1:1500) |
| Antibody | SDHA | self made | PRAB4978 | (1:2000) |
| Antibody | Rieske | self made | PRAB1512 | (1:2000) |
| Antibody | ATP5B | self made | PRAB4826 | (1:10000) |
| Antibody | TIM44 | Proteintech | RRID:AB_2204679 | (1:2500) |
| Antibody | NDUFB8 | self made | PRAB3764 | (1:500) |
| Antibody | NDUFA9 | self made | PRAB1524 | (1:500) |

*Continued on next page*

*Continued*

| Reagent type (species) or resource | Designation | Source or reference | Identifiers | Additional information |
|---|---|---|---|---|
| Antibody | LETM1 | self made | PRAB538 | (1:5000) |
| Antibody | TIM23 | self made | PRAB1527 | (1:2000) |
| Antibody | SCO1 | self made | PRAB4980 | (1:500) |
| Antibody | COA6 | self made | PRAB5007 | (1:500) |
| Antibody | COX6C | self made | PRAB4950 | (1:2000) |
| Antibody | SCO2 | self made | PRAB4982 | (1:500) |
| Antibody | FAM36A | self made | PRAB4490 | (1:500) |
| Antibody | SURF1 | self made | PRAB1528 | (1:1000) |
| Recombinant DNA reagent | QuikChange Site-Directed Mutagenesis Kit | Agilent | 210515 | |
| Recombinant DNA reagent | KOD Hot Start DNA Polymerase | Merck | 71086–3 | |
| Recombinant DNA reagent | First Strand cDNA Synthesis kit | ThermoFisher Scientific | K1612 | |
| Commercial assay or kit | Human Complex IV activity kit | Abcam | ab109910 | |
| Chemical compound, drug | GeneJuice | Merck | 70967–3 | |
| Chemical compound, drug | Anti-FLAG M2 Affinity Gel | Sigma-Aldrich | A2220 | |
| Chemical compound, drug | Trizol | ThermoFisher Scientific | 15596026 | |
| Chemical compound, drug | Protein-A SepharoseTM CL-4B | GE Healthcare | 17-0963-03 | |
| Chemical compound, drug | [35S]methionine | Hartmann Analytic | SCM-01 | |
| Chemical compound, drug | Emetine dihydrochloride hydrate | Sigma-Aldrich | 219282 | |
| Chemical compound, drug | Anisomycin | AppliChem | A7650,0025 | |
| Software, algorithm | ImageQuantTL 7.0 software | GE Healthcare | RRID:SCR_014246 | |
| Software, algorithm | ImageJ 1.47 v | NIH | RRID:SCR_003070 | |
| Software, algorithm | Geneious | Biomatters Ltd | RRID:SCR_010519 | |
| Software, algorithm | Prism5 | GraphPad Software | RRID:SCR_015807 | |

## Cell culture

HEK Flp-In T-REx 293 (Invitrogen, Carlsbad, CA) (HEK-293T) were cultured in DMEM, supplemented with 10% [v/v] fetal bovine serum (FBS) (GIBCO, Invitrogen), 2 mM L-glutamine and 50 µg/ml uridine at 37°C under a 5% $CO_2$ humidified atmosphere. The cell lines were authenticated by STR profiling using eight different and highly polymorphic short tandem repeat loci at the Leibniz-Institut DSMZ, Braunschweig, Germany. All cell lines were regularly monitored for mycoplasma. Cell were treated either with 20 µg/ml emetine (Sigma-Aldrich GmbH, Munich, Germany) for 6 hr or with 50 µg/ml thiamphenicol (Sigma-Aldrich) for 2 days in DMEM medium, for inhibition of cytosolic or mitochondrial translation, respectively. Transfections were performed according to manufacturer's recommendations using GeneJuice (Novagen, Merck KGaA, Darmstadt, Germany). Briefly, approximately 300,000 cells/25 $cm^2$ were transfected using 4 µl of transfection reagent and 1 µg of DNA. Cells were either harvested or subjected to drug selection, 48 hr after transfections. COX16$^{-/-}$ HEK-293T cell line was generated applying the CRISPR/Cas9 technology as previously described (*Ran et al., 2013*; *Richter-Dennerlein et al., 2016*). Briefly, oligonucleotides GCGAAAAGCACGCATCACCG and CGGTGATGCGTGCTTTTCGC containing the guide sequences were annealed and ligated into

the pX330 vector. HEK-293T cells were co-transfected with pX330 and with the pEGFP-N1 plasmid. After three days, single cells expressing GFP were sorted by flowcytometry into 96-well plates. After colony expansion, single colonies were screened by immunoblotting.

## Mitochondrial isolation and protein localization assays

Isolation of mitochondria from cultured cells was performed as described previously (*Dennerlein et al., 2015*; *Richter-Dennerlein et al., 2016*). Bradford analysis using BSA as a standard was used to measure protein concentrations. Carbonate extraction and mitochondrial swelling experiments were implemented as described previously (*Fiumera et al., 2007*; *Mick et al., 2012*; *Soto et al., 2012*). Briefly, for carbonate extraction of proteins, isolated mitochondria were incubated in 100 mM Na2CO3 (pH 10.8 or 11.5) followed by centrifugation for 30 min at 100,000 x g at 4°C. Analysis of submitochondrial localization of proteins was performed by protease protection assay using proteinase K (PK). Isolated mitochondria were resuspended either in SEM buffer (250 mM sucrose, 1 mM EDTA, and 10 mM MOPS [pH 7.2]), to osmotically stabilize them, or in EM buffer (1 mM EDTA, and 10 mM MOPS [pH 7.2]), to rupture the outer mitochondrial membrane. As a positive control, mitochondrial membranes disrupted either by 1% Triton X-100 for carbonate extraction experiments or by sonication for submitochondrial localization experiments.

## In vivo labeling of mitochondrial translation products with [$^{35}$S] methionine

In vivo labeling in human cells was performed as described previously (*Chomyn, 1996*; *Fiumera et al., 2009*; *Khalimonchuk and Winge, 2008*). Inhibition of cytosolic translation was achieved by treating cells either with 100 µg/ml emetine during pulse experiments, or with 100 µg/ml anisomycin (Sigma-Aldrich) in pulse chase experiments. Mitochondrial translation products were labeled with 0.2 mCi/ml [$^{35}$S]methionine for 1 hr. For chase experiments, the radioactive medium was substituted by adding fresh growth medium, followed by incubation at 37°C under 5% $CO_2$ for the indicated time points. The cells were harvested in 1 mM EDTA/PBS. The samples were further analyzed either by SDS- or BN-PAGE and processed for affinity purification procedures. Signals were obtained by auto-radiography.

## BN-PAGE analysis

Standard protocol for BN-PAGE analysis was followed as previously described (*Baertling et al., 2015*; *Ghosh et al., 2016*; *Mick et al., 2012*; *Pacheu-Grau et al., 2015*; *Stroud et al., 2015*). In short, to separate native protein complexes, whole cells or isolated mitochondria were incubated in lysis buffer (1% digitonin, 50 mM Tris-HCl [pH 7.4], 20 mM MgCl2, 50 mM NaCl, 10% glycerol, and 1 mM PMSF). To remove the non-solubilized material the lysate was centrifuged at 20,000 xg for 15 min at 4°C. The supernatant was mixed with BN-loading buffer (0.5% Coomassie Brilliant Blue G-250, 50 mM 6-aminocaproic acid, 10 mM Bis-Tris/HCl [pH 7]) and subjected to BN-PAGE analysis. In-gel activity assays were performed according to published procedures (*Bourens et al., 2014*; *Wittig et al., 2006*).

## Cytochrome C Oxidase activity and quantitation assay

Specific activity and relative amount of cytochrome *c* oxidase were determined according to the manufacturer's instructions using Complex IV Human Specific Activity Microplate Assay Kit (Mitosciences, Abcam, Cambridge, United Kingdom). Total 15 µg of cell lysate was loaded per well. Cytochrome *c* oxidase activity was calculated by measuring the oxidization of cytochrome *c* and the decrease of absorbance at 550 nm. To measure the relative COX amounts, the lysates from the same batch were incubated with a specific cytochrome *c* oxidase antibody, conjugated to alkaline phosphatase. The increase of absorbance at 405 nm was measured.

## Affinity purification procedures

Whole cells or isolated mitochondria (0.5 mg) were resuspended in lysis-buffer (50 mM Tris-HCl [pH 7.4], 150 mM NaCl, 0.1 mM EDTA, 10% glycerol, 1 mM PMSF, and 1% digitonin) to a final concentration of 1 mg/ml. This was followed by an incubation at 30 min at 4°C under mild agitation. Non-solubilized material was removed by centrifugation at 20,000 xg, 4°C for 15 min. Supernatants were incubated with anti-FLAG-agarose (Sigma-Aldrich) or ProteinA-Sepharose (GE

Healthcare, Chicago, IL) conjugated with specific or control antibodies. After washing of the resin, proteins were eluted with FLAG peptides or by pH shift (0.1 M Glycin [pH 2.8]).

## Miscellaneous

SDS-PAGE and western-blotting of proteins to PVDF membranes (Millipore, Merck KGaA, Darmstadt, Germany) was performed using standard methods. Primary antibodies were raised in rabbits or purchased (anti-COX16, Protientech). HRP-coupled secondary antibodies applied to antigen–antibody complexes and detected by enhanced chemiluminescence on X-ray films.

## Statistical analyses

Data are expressed as mean ± SEM. Significant differences between groups were analyzed using Prism five software (GraphPad Software, San Diego, CA) by unpaired Student $t$ test and ANOVA, unless otherwise noted.

## Acknowledgements

We thank A Barrientos for the FAM36A$^{-/-}$ cell line. Supported by the Deutsche Forschungsgemeinschaft SFB1002 (PR), the European Research Council (ERC) Advanced Grant (ERCAdG No. 339580) to PR, MWK FoP 88b (PR), and the Max Planck Society (PR).

## Additional information

### Funding

| Funder | Grant reference number | Author |
|---|---|---|
| H2020 European Research Council | ERCAdG No. 339580 | Peter Rehling |
| Deutsche Forschungsgemeinschaft | SFB1002 | Peter Rehling |

The funders had no role in study design, data collection and interpretation, or the decision to submit the work for publication.

### Author contributions

Abhishek Aich, Conceptualization, Formal analysis, Validation, Investigation, Visualization, Methodology, Writing—original draft, Writing—review and editing; Cong Wang, Formal analysis, Validation, Investigation, Visualization; Arpita Chowdhury, Validation, Investigation, Visualization, Writing—review and editing; Christin Ronsör, Formal analysis, Validation, Methodology; David Pacheu-Grau, Validation, Investigation, Methodology; Ricarda Richter-Dennerlein, Conceptualization, Formal analysis, Supervision, Validation, Investigation, Visualization, Methodology, Writing—original draft; Sven Dennerlein, Conceptualization, Data curation, Formal analysis, Supervision, Validation, Investigation, Visualization, Methodology, Writing—original draft, Project administration, Writing—review and editing; Peter Rehling, Conceptualization, Resources, Formal analysis, Supervision, Funding acquisition, Visualization, Writing—original draft, Project administration, Writing—review and editing

### Author ORCIDs

Abhishek Aich http://orcid.org/0000-0001-8331-9874
Peter Rehling http://orcid.org/0000-0001-5661-5272

### Decision letter and Author response

Decision letter https://doi.org/10.7554/eLife.32572.010
Author response https://doi.org/10.7554/eLife.32572.011

## Additional files

**Supplementary files**

• Transparent reporting form
DOI: https://doi.org/10.7554/eLife.32572.008

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
