## [Decision Letter]

Thank you for submitting your article "COX16 merges COX1 and COX2 assembly pathways promoting COX2 metalation" for consideration by *eLife*. Your article has been reviewed by three peer reviewers, and the evaluation has been overseen by a Reviewing Editor and Anna Akhmanova as the Senior Editor. The reviewers have opted to remain anonymous.

The reviewers have discussed the reviews with one another and the Reviewing Editor has drafted this decision to help you prepare a revised submission.

Summary:

This manuscript describes the function of the mitochondrial protein COX16 in human cytochrome c oxidase (COX) biogenesis. COX16 was originally described in yeast as a COX assembly factor but its specific role remains uncharacterized. From yeast to humans, COX biogenesis involves the formation of modules around the core subunits, which eventually come together. The two best-studied modules are those leading to COX1 or COX2 maturation and assembly. COX1 and COX2 are stabilized by subunit-specific chaperones that facilitate or promote the formation of the copper or heme redox centers that the two proteins contain. The current manuscript clearly shows that COX16 promotes COX2 metallation and that plays an important role in the COX2 assembly line. The meaning of the interaction with COX1 module components is interpreted as a way to promote its merging with the COX2 module.

Essential revisions:

1) In the subsection “COX16 is required for cytochrome *c* oxidase biogenesis”, it is mentioned that COX16 mutant mitochondria displayed a marked reduction in the levels of COX1 and COX2. Whereas this is true for COX2, the levels of COX1 closer to WT levels. In fact, as a result of COX1 stability in the absence of COX assembly, MITRAC subcomplexes accumulate in the absence of COX16, as seen in Figure 2. The conclusion needs to be modified.

2) Subsection “COX16 is required for COX2 assembly”: the authors explain that COX2 is markedly unstable in COX16 mutant than in WT cells. However, this is not seen in Figure 3, where COX2 and COX3 run together. In Figure 3, some statistics showing whether or not the differences are significant, time-point by time-point would be helpful. Alternatively, the authors should down-tune the statement on COX2 stability, as the effect seems to be minor.

3) The protein synthesis experiment in Figure 3, clearly shows an enhanced synthesis and stability of ATP6 and ATP8 in the absence of COX16. This has been seen in other COX assembly mutants but nonetheless should be mentioned and discussed in the text. Also, the authors should comment why they think that CI activity is increased in the COX16 mutant cell line (Figure 2).

4) Subsection “COX16 is required for COX2 assembly”: The authors claim that 'the ratios of COX1 present in the mature monomeric protein complex IV and in MITRAC complexes changed significantly in absence of COX16'. This is again a strong statement given the rather subtle changes presented in Figure 3. This needs to be down-tuned or more convincing evidence needs to be presented.

5) Figure 3 indicates that in the COX16^-/-^ cells, COX1 populates the CIV and MITRAC complexes. The slightly increased relative amounts of MITRAC-COX1 in the mutant suggests that the assembly is slower or weaker, however, considerable levels of COX1 still reach the CIV complex stage. According to this experiment, the major difference is that there is basically no COX1 in supercomplexes. Doesn't this suggest that COX16 is critical for supercomplex formation rather than for cytochrome oxidase assembly? Along the same lines: in the blot shown in Figure 3, COX2 seems to be present both in WT and in COX16^-/-^ cells. Again, this suggests that COX16 is not particularly important for cytochrome oxidase assembly but rather increases – directly or indirectly – to some degree the efficiency of COX assembly.

6) COX16 is shown to interact only with MITRAC12 but not with any other MITRAC complexes component, including COX1. This is surprising, considering that MITRAC12 interacts with the nascent COX1 polypeptide, almost at the same time as C12orf62. One way to explain these observations is that there are two pools of MITRAC12, one of which would actually interact with COX16 or the COX16-containing COX2 module, to help it merge with the COX1 module. In this model, it would be MITRAC12 and not COX16 the COX chaperone playing the merging role during COX1-COX2 assembly. The authors should try to distinguish between these two possibilities. Furthermore, results presented in Figure 1 and Figure 3 are apparently contradictory regarding the interaction of COX16 with MITRAC12. Some explanation for the discrepancy is required.

7) In their model the authors suggest that COA6 is part of an intermediate that is 'earlier' than the COX16-containing module. However, in Figure 4, they show convincingly that COX16 interacts with COA6. This does not fit together. Doesn't this indicate that COA6 and COX16 are present in the same assembly intermediate? The authors should at least comment on this.

8) Please change the title of the paper to make it more broadly accessible.

---

## [Author Response]

Essential revisions:1) In the subsection “COX16 is required for cytochrome c oxidase biogenesis”, it is mentioned that COX16 mutant mitochondria displayed a marked reduction in the levels of COX1 and COX2. Whereas this is true for COX2, the levels of COX1 closer to WT levels. In fact, as a result of COX1 stability in the absence of COX assembly, MITRAC subcomplexes accumulate in the absence of COX16, as seen in Figure 2. The conclusion needs to be modified.

As suggested, we modified the statement.

2) Subsection “COX16 is required for COX2 assembly”: the authors explain that COX2 is markedly unstable in COX16 mutant than in WT cells. However, this is not seen in Figure 3, where COX2 and COX3 run together. In Figure 3, some statistics showing whether or not the differences are significant, time-point by time-point would be helpful. Alternatively, the authors should down-tune the statement on COX2 stability, as the effect seems to be minor.

As requested, we add p values to the analysis in Figure 3 and provide a corresponding description in the figure legend.

3) The protein synthesis experiment in Figure 3, clearly shows an enhanced synthesis and stability of ATP6 and ATP8 in the absence of COX16. This has been seen in other COX assembly mutants but nonetheless should be mentioned and discussed in the text. Also, the authors should comment why they think that CI activity is increased in the COX16 mutant cell line (Figure 2).

As requested, we incorporated the observation on the increase in ATP6 and ATP8. Moreover, we comment on the surprising increase in CI activity in the COX16 mutant cell line for which we have no explanation.

4) Subsection “COX16 is required for COX2 assembly”: The authors claim that 'the ratios of COX1 present in the mature monomeric protein complex IV and in MITRAC complexes changed significantly in absence of COX16'. This is again a strong statement given the rather subtle changes presented in Figure 3. This needs to be down-tuned or more convincing evidence needs to be presented.

To address the point raised by the reviewer, we provide new experimental data demonstrating that upon use of the detergent DDM, COX1 is mostly in MITRAC complexes in the absence of COX16, (Figure 2). We would also like to point out, that upon immunoisolation of MITRAC intermediates in digitonin buffer (through C12ORF62 or MITRAC12) more COX1 is recovered than in the wild type sample (see Figure 5). These findings support the conclusion, that increased amounts of COX1 accumulate in MITRAC in the absence of COX16.

5) Figure 3 indicates that in the COX16^-/-^ cells, COX1 populates the CIV and MITRAC complexes. The slightly increased relative amounts of MITRAC-COX1 in the mutant suggests that the assembly is slower or weaker, however, considerable levels of COX1 still reach the CIV complex stage. According to this experiment, the major difference is that there is basically no COX1 in supercomplexes. Doesn't this suggest that COX16 is critical for supercomplex formation rather than for cytochrome oxidase assembly? Along the same lines: in the blot shown in Figure 3, COX2 seems to be present both in WT and in COX16^-/-^ cells. Again, this suggests that COX16 is not particularly important for cytochrome oxidase assembly but rather increases – directly or indirectly – to some degree the efficiency of COX assembly.

To address this point directly, we analyzed the amount of mature complex IV upon DDM solubilization (see point 4). Apparently, there is very little mature complex IV in COX16 mutant cells (Figure 2). Moreover, to assess if COX16 was involved in supercomplex formation, we analyzed if COX16 was associated with supercomplexes. This was clearly not the case as shown in the new Figure 4. Hence, differences in supercomplex organization seen in the mutant are apparently due to the loss of complex IV and a concomitant reduction of complex IV-containing supercomplexes. One explanation, that we do see remaining complex IV in supercomplexes could be a different half time for monomeric complex IV compared to supercomplex-associated complex IV. However, as in gel activity assays and Western blot BN-PAGE analyses are not reliable quantitative assays, we believe that the activity assay presented in Figure 2 provides the best support for a reduction in mature complex IV, by approx. 50%. We feel that this clearly demonstrates a role in complex IV biogenesis but do agree that COX16 rather promotes complex IV biogenesis and is not essential. In fact, many complex IV assembly factors display bypass assembly to different extends. We incorporated these considerations into the revised manuscript and into the new title.

6) COX16 is shown to interact only with MITRAC12 but not with any other MITRAC complexes component, including COX1. This is surprising, considering that MITRAC12 interacts with the nascent COX1 polypeptide, almost at the same time as C12orf62. One way to explain these observations is that there are two pools of MITRAC12, one of which would actually interact with COX16 or the COX16-containing COX2 module, to help it merge with the COX1 module. In this model, it would be MITRAC12 and not COX16 the COX chaperone playing the merging role during COX1-COX2 assembly. The authors should try to distinguish between these two possibilities. Furthermore, results presented in Figure 1 and Figure 3 are apparently contradictory regarding the interaction of COX16 with MITRAC12. Some explanation for the discrepancy is required.

Figure 5 show that the amount of COX16, that copurifies with MITRAC12 is drastically less than what is found associated with COA6 or FAM36A. This could reflect a transient interaction between MITRAC and COX16. To support MITRAC and COX1 association of COX16, we purified C12ORF62-containing complexes with C12ORF62^FLAG^ as suggested. Under these conditions we were able to detect COX16 in the eluate (new Figure 5). The observed specific association of MITRAC12 with COX2, could indeed reflect a second pool of MITRAC12 especially since C12orf62 does not pull down newly synthesized COX2. However, the magnitude of MITRAC12 association with COX2 depends on COX16. We discuss this in the text and changed the title.

7) In their model the authors suggest that COA6 is part of an intermediate that is 'earlier' than the COX16-containing module. However, in Figure 4, they show convincingly that COX16 interacts with COA6. This does not fit together. Doesn't this indicate that COA6 and COX16 are present in the same assembly intermediate? The authors should at least comment on this.

We thank the reviewer for pointing this error out. As proposed, we have corrected the model in Figure 6.

8) Please change the title of the paper to make it more broadly accessible.

We modified the title as suggested.